# Physical Treatments to Control *Clostridium botulinum* Hazards in Food

**DOI:** 10.3390/foods12081580

**Published:** 2023-04-07

**Authors:** Muhammad Tanveer Munir, Narjes Mtimet, Laurent Guillier, François Meurens, Phillipe Fravalo, Michel Federighi, Pauline Kooh

**Affiliations:** 1EnvA, Unit of Hygiene, Quality and Food Safety, 94700 Maisons-Alfort, France; muhammad.munir@vet-alfort.fr (M.T.M.); narjes.mtimet@vet-alfort.fr (N.M.); 2Anses, Laboratory of Food Safety, 94700 Maisons-Alfort, France; 3Anses, Unit UERALIM, 94700 Maisons-Alfort, France; laurent.guillier@anses.fr (L.G.); pauline.kooh@anses.fr (P.K.); 4INRAE, Oniris, BIOEPAR, 44307 Nantes, France; francois.meurens@oniris-nantes.fr; 5Swine and Poultry Infectious Diseases Research Center, Faculty of Veterinary Medicine, University of Montreal, St-Hyacinthe, QC J2S 2M2, Canada; 6Chaire Agroalimentaire du Cnam, Conservatoire des Arts et Métiers, EPN7, 22440 Ploufragan, France; philippe.fravalo@lecnam.net

**Keywords:** botulinum, control, food, physical treatments, prevention

## Abstract

*Clostridium botulinum* produces Botulinum neurotoxins (BoNTs), causing a rare but potentially deadly type of food poisoning called foodborne botulism. This review aims to provide information on the bacterium, spores, toxins, and botulisms, and describe the use of physical treatments (e.g., heating, pressure, irradiation, and other emerging technologies) to control this biological hazard in food. As the spores of this bacterium can resist various harsh environmental conditions, such as high temperatures, the thermal inactivation of 12-log of *C. botulinum* type A spores remains the standard for the commercial sterilization of food products. However, recent advancements in non-thermal physical treatments present an alternative to thermal sterilization with some limitations. Low- (<2 kGy) and medium (3–5 kGy)-dose ionizing irradiations are effective for a log reduction of vegetative cells and spores, respectively; however, very high doses (>10 kGy) are required to inactivate BoNTs. High-pressure processing (HPP), even at 1.5 GPa, does not inactivate the spores and requires heat combination to achieve its goal. Other emerging technologies have also shown some promise against vegetative cells and spores; however, their application to *C. botulinum* is very limited. Various factors related to bacteria (e.g., vegetative stage, growth conditions, injury status, type of bacteria, etc.) food matrix (e.g., compositions, state, pH, temperature, aw, etc.), and the method (e.g., power, energy, frequency, distance from the source to target, etc.) influence the efficacy of these treatments against *C. botulinum*. Moreover, the mode of action of different physical technologies is different, which provides an opportunity to combine different physical treatment methods in order to achieve additive and/or synergistic effects. This review is intended to guide the decision-makers, researchers, and educators in using physical treatments to control *C. botulinum* hazards.

## 1. Introduction

*Clostridium botulinum* is a Gram-positive, spore-forming bacterium with ubiquitous distribution in the environment. It produces botulinum neurotoxin (BoNT), which is responsible for botulism (mainly characterized by flaccid paralysis) in humans and animals [1]. The majority of cases in humans are foodborne botulism [2], therefore, it is a crucial food hazard [3].

Various physical (e.g., heating and irradiation) [4], chemical (e.g., nitrate, nitrite, organic acid, and natural antimicrobial compounds) [5,6], and biological (e.g., bacteriophages and enzymes) [7,8] treatments are used to control biological hazards in food. For example, nitrite is a chemical additive used for curing meat products against vegetative *C. botulinum*; however, this practice is being questioned, because nitrite has carcinogenic effects [9]. As the spores of *C. botulinum* are very resistant to environmental conditions, biological and chemical treatments are not very effective against them and can even elicit secondary food safety concerns [5]; therefore, physical treatments are the main choice to control *C. botulinum* hazards including bacterium, spores, and toxins. In fact, *C. botulinum* spores are one of the most heat-resistant spores among pathogenic microorganisms [3]. Therefore, their inactivation is considered a standard for thermally processed, commercially sterile food products, and the awaited 12-log reduction is named “Botulinum cook” [3].

Among physical food treatments, heating is the principal and traditional method of microbial inactivation. Furthermore, the spores of *C. botulinum* are the reference point for establishing thermal treatment efficiency scales for “low-acid canned foods” [10]. These heat treatment benefits come with some undesired effects on food, though, such as changes in the physicochemical properties and organoleptic characteristics. Therefore, alternative technologies have always been in demand in order to conserve these product properties while inactivating the *C. botulinum* hazard [4]. Ionizing radiation was tested for food safety in the 1960s [4,11], and since then, there have been many recent advances in other nonthermal technologies, such as high-pressure processing (HPP), cold plasma (CP), pulsed electric field (PEF), intense light pulses (ILP), ultraviolet (UV), ultrasound waves, etc. Understanding and analyzing the specific mode of action of different technologies aids in the design and implementation of strategies for exploiting their potential cumulative or synergistic effects in order to control the *C. botulinum* hazard in food products [12,13,14,15,16,17,18,19].

This literature review intends to provide a brief overview of the current state of knowledge regarding *C. botulinum* (cells, spores, neurotoxin, and botulism), and the common physical treatments (along with their mode of action) to eliminate this biological hazard in the food industry. This article aims at providing basic knowledge for educational purposes, giving relevant information to food industry professionals about the available control options, and highlighting the knowledge gaps and research directions for researchers in food safety.

## 2. *Clostridium botulinum* Cells, Spores, Toxins, and the Disease

### 2.1. The Bacterium

*Clostridium botulinum* is an anaerobic, straight or slightly curved, motile rod (3–22 μm by 0.2–0.6 μm in size), Gram-positive, catalase-negative, and mainly sub-terminal to rarely central spore-forming bacterium [3,20,21]. The genus *Clostridium* is a member of the family Clostridiaceae, in the order Clostridiales, class Clostridia, and phylum Bacillota (synonym Firmicutes) [22]. This genus consists of approximately 200 species, about 15 of which synthesize toxins that cause disease in humans or animals [3,23].

Based on biochemical and proteolytical properties, *C. botulinum* is divided into three basic groups (I–III). Strains of Group I are proteolytic, and strains of groups II and III are non-proteolytic [22]. This classification of *C. botulinum* has also been confirmed by genetic studies based on the comparison of DNA sequences and their level of homology as well as the sequences of the gene encoding the 16S rRNA of the different clostridia (Table 1). For the sake of thoroughness, it is important to note that other species of the genus *Clostridium* than *C. botulinum,* such as *C. baratii,* also produce botulinum toxins and constitute groups IV, V, and VI (not shown in Table 1) of botulinum toxin-producing *Clostridium*. Group IV is proteolytic and groups V and VI are non-proteolytic [22,24,25].

Neurotoxigenic *Clostridium* strains are also classified according to the type of toxin they produce. Currently, nine types of BoNTs have been identified: A, B, C, D, E, F, G, H, and X [26], but only the first seven (A to G) are considered for this classification [25,27]. Table 1 gives the correspondence between groups I to III, and the types of toxins produced [28]. This table also shows the species of *Clostridium* related to *C. botulinum*. These species are the bacteria that have a similar biochemical and proteolytic profile to *C. botulinum* but do not produce botulinum toxin.

In general, the most commonly used classification is related to the type of toxin produced. This approach also considers the fact that different strains of *C. botulinum* are capable of producing two or more types of BoNTs simultaneously [29], usually with different rates of production of the different toxins (Table 1). In the literature, by convention, the type of toxin produced in the majority is indicated in capital letters and the type of toxin produced in the minority is indicated in lowercase [30], e.g., Ba, Bf, Ab, Af, Bx, and Bh [23,31].

### 2.2. Spores

Sporulation in *C. botulinum* is a survival and pathogenesis strategy and forms oval-shaped spores (Figure 1). Under stress conditions, the stress response regulator Spo0A initiates spore formation by activating or repressing the expression of genes encoding the early sporulation regulators [32]. A vegetative cell goes into an asymmetrical septation phase, producing a mother cell and a forespore. This forespore is engulfed by the mother cell forming the protective layers [33]. The spore matures inside this cell and finally released as a dormant spore by rupturing the mother cell. *C. botulinum* spores are one of the most heat-resistant spores among pathogenic organisms; therefore, their inactivation is considered the standard of commercial sterilization. In the absence of oxygen, and other suitable conditions, the spores germinate, grow, and then excrete toxins [1].

Suitable environmental conditions (nutrient and non-nutrient germinants) activate the germinant receptor located in the spore inner membrane in order to initiate the process of germination. In the first stage, dipicolinic acid (DPA) is released and partial core hydration occurs. Subsequently, the cortex–lytic enzymes cause the degradation of the cortex and the peptidoglycan is hydrolyzed. Ultimately, the core swelling and hydration are followed by cell outgrowth [34,35].

### 2.3. BoNTs

The sporulation and germination in *C. botulinum* produce exotoxins including neurotoxins [32]. The BoNT is one of the most potent known biological toxins, therefore this toxin is included among the potential bioterrorist threats [1,22,28,36]. In vitro, the highest levels of BoNTare produced at the end of the exponential growth and in the early stationary growth phases [37].

The BoNTs share a similar structure to a protoxin of 150 kD protein, having two chains covalently bound by interchain disulfide bonds [38]. It cleaves into an active toxin consisting of a heavy (100 kD) and a light (50 kD) chain [39]. In proteolytic strains, protease is responsible for this activation of toxin while in the case of non-proteolytic strains, the toxin is activated by the influence of other proteolytic organisms present in the culture [1,38].

The BoNTs are divided into antigenically distinguishable types (i.e., A, B, C1, C2, D, E, F, and G), as mentioned in the previous section [22,38]. The BoNTs A, B, E, and F are produced by strains of groups I and II and they are the principal cause of botulism in humans. The BoNT types C, D, and E cause illness in other mammals, birds, and fish [3].

### 2.4. Botulism

Botulism is a severe paralytic disease caused by BoNT. This illness has different forms based on the mode of contamination and exposure to the BoNTs (Table 2). Foodborne botulism occurs after ingesting BoNT-containing foods (canned vegetables—A, preserved meat—B, and fish—E) [1]. Inhalation botulism has a similar clinical footprint to food botulism, and it occurs by inhaling an aerosol that is accidentally or intentionally (e.g., bioterrorism) contaminated with BoNTs [1]. Wound botulism occurs as a result of injecting *C. botulinum*-spore contaminated drugs into the skin or muscles, or direct contamination of the wound during a traumatic injury [28]. Once ingested, *C. botulinum* can also colonize the intestinal tract and induce infant botulism or adult intestinal toxemia [36].

Botulism is a rare but deadly illness. Extrapolation from animal models shows that the median lethal dose of BoNT for an adult male human is estimated to be 5 ng by the parenteral route [22], 30 ng by the oral route, and 130 ng by inhalation [1,40]. In Europe, about 100 confirmed cases are reported every year (82 in 2020) with the highest number of cases being reported in Italy, France, Poland, and Romania [41]. In the US, on average, 110 botulinum cases occur annually, and 25% of these are food-borne botulism [42].

The incubation period of *C. botulinum* ranges from 1 to 10 days, and in the majority of cases, it is just 1 to 3 days (Table 2). The duration of botulism symptoms ranges from a few days to 8 months. Full recovery may take months or even years [22].

Different BoNTs have a similar mode of action when causing botulism [38]. BoNT does not cross the blood–brain barrier and thus affects only motor nerves. Here, the heavy chain of activated toxin allows for endocytosis of the low chain into the peripheral neuron cell, which then binds to the soluble N-ethylmaleimide-sensitive factor attachment protein receptor (SNARE) proteins. These proteins are responsible for the exocytosis of acetylcholine synaptosomes, and they are destroyed by this toxin binding [20,27]. Consequently, the acetylcholine is not released and the nerve activity used to control the muscles is lost [36].

**Table 2 foods-12-01580-t002:** Incubation time and clinical overview of the main types of human botulisms *.

Types of Botulism	Incubation	Clinical Overview
Foodborne and inhalation botulism	12–72 h after ingestion or inhalation (minimum 6 hours) can be up to 10 days	Diplopia, nausea, vomiting, abdominal pain, diarrhea Subsequently, cranial nerve damage and descending paralysis The case fatality rate is <4%
Infant botulism	Not precisely determined, but may be short (less than three days)	Constipation at first, followed by a lethargic appearance, feeding difficulties, descending or generalized hypotonia, anorexia, irritability, weakened voice (including impaired crying) Muscular paralysis in very severe forms The case fatality rate is <1%
Wound botulism	4–17 days after injection or traumatic injury	No gastrointestinal symptoms Presence of signs and symptoms related to cranial nerve damage and descending muscle paralysis The case fatality rate is 5–10%

* Adapted from [1,3,42].

### 2.5. Stability of C. botulinum, and Their Spores and Toxins

The bacterium, spores, and BoNT show different levels of stability in different environmental conditions (Table 3). Likewise, the control of this biological hazard requires various intensities of physical treatments for denaturing the toxin and inactivating the spores and vegetative cells depending upon the food matrices and food safety objectives.

Overall, *C. botulinum* is a rare but extremely severe pathogen. Its transmission through food is now proven in all its forms: vegetative, spores, and toxins [1,22,28]. These characteristics make it imperative to be able to control all the different forms of this hazard during food production and until consumption. As with all foodborne biological hazards, the control strategy can be preventive and/or curative [49]. In terms of curative strategies, physical food processing is an excellent means of control, because it is reliable, effective, predictable, and measurable. These different properties mean that they are often associated with critical control steps in HACCP plans. In the next part of this article, we will review these main processes.

It is also important to mention that the classic studies on *C. botulinum* are very old, because this microorganism is now declared a potential bioterrorism threat and there are only a few laboratories authorized to work on this bacterium. This is the reason that thermal treatments are well-studied for this microorganism. Meanwhile, most of the ionizing radiation research regarding *C. botulinum* was conducted in the 1950–60s. Moreover, there are not many publications on the application of alternative physical treatments to control the *C. botulinum* hazard. Therefore, some of the results in this article are extrapolated from surrogate or model bacteria.

## 3. Thermal Treatments against Botulinum Hazard

Exposure of *C. botulinum* to temperatures beyond a certain threshold results in the denaturation and aggregation of cellular proteins, leading to the loss of numerous enzymatic functions. Thermal energy also changes the permeability of membranes and denatures cytosol components including RNA and ribosomes. Although DNA is comparatively more resistant to heat damage, its function depends upon the availability of interlinked enzymes [50].

Heating damages *C. botulinum* vegetative cells depending on various factors before, during, or after the treatment. Before the treatment, the stage of growth can influence the bacterial cells’ resistance to heat, e.g., exponentially growing cells are usually more sensitive to heat than cells in the stationary phase of growth [51]. Exposure to different sublethal stresses such as growth temperature and media characteristics (type, composition, pH, and aw), before or during treatment, activates the resistance mechanisms in bacteria which can contribute to cross-resistance and increase the thermotolerance of bacterial cells [52]. During treatment, the heat intensity and the characteristics of the treatment medium influence the thermosensitivity of bacteria [53]. Odlaug and Pflug, [54] reported that the D-values (decimal reduction time) of *C. botulinum* type A and type B strains were approximately three times higher in the phosphate buffer (0.07 M, pH 7.0) than in tomato juice (pH 4.2). Overall, heating causes sublethal or lethal damage to cells, and their recovery from heat stress depends upon several factors such as the intensity of treatment, the treatment media, the physiological state of *C. botulinum*, and the recovery conditions [51].

The thermotolerance of *C. botulinum* varies both between and within groups. The proteolytic strains of Group I optimally grow at warmer temperatures compared to the non-proteolytic strains of Group II (Table 3). Generally, the pasteurization of food allows for the destruction of *C. botulinum* vegetative cells. At sublethal doses, the cells start producing spores and may overcome low heating by activating their cell repair mechanisms; however, they may lose their ability to produce toxins in this remodeling process [55].

The *C. botulinum* spores survive mild heat treatments and extended shelf life of food products, even at low temperatures, can allow spore germination, outgrowth, and toxigenesis [56,57]. For example, Pernu et al., [58] reported a high prevalence (32%) of *C. botulinum* in mild heat-treated, vacuum-packaged vegetarian sausages. As some of the strains, e.g., group II, can produce toxins even at refrigeration temperatures, the presence of spores poses a high risk of *C. botulinum* growth and toxin production [58]. It is generally accepted that the D_121_ °C for most resistant spores (*C. botulinum* A) is 0.21 min in low-acid canned food and heating up to 3 min (botulinum cook) allows at least 12-log reductions (commercial sterility). It is important to consider that the ability of spores to resist heat and recover from thermal stress can be influenced by a variety of factors, including the environment, strain, thickness of the spore cortex, and properties of the food matrix (Table 3) [53,59,60,61,62,63]. Marshall et al. [62] investigated the effect of sporulation temperature on the heat resistance of *C. Botulinum* type A spores. The maximal heat resistance (D_100 °C_) was obtained from spore suspensions produced at a sporulation temperature close to the growth optimum (i.e., D_100 °C_ at a sporulation temperature of 20 °C = 3.4 min, at 27 °C = 5.08 min, and at 37 °C = 5.65 min). Several studies have performed a meta-analysis of thermal destruction data and fitted secondary models to predict the D-value as a function of temperature for proteolytic and non-proteolytic forms, taking into account strain variability [46,47,64].

Heating can also inactivate the BoNTs depending on various factors, such as the medium composition, strain type, treatment temperature, etc. Bradshaw et al. [65] observed that at temperatures from 68–80 °C, the BoNT inactivation times in mushroom patties ranged from 53.15 min to 0.62 min in type A (73A) and 51.20 min to 1.08 min in type B (Beans-B) toxins. They also observed that a significantly shorter toxin inactivation time was required in phosphate buffer. Different types of BoNTs have different stability to heat, and some can even survive common pasteurization. Rasooly and Do [66] reported that low-heat milk pasteurization (63 °C, 30 min) inactivated BoNT of serotype A but not of serotype B. Therefore, high-temperature pasteurization should be used to inactivate all the BoNTs and minimize the risk of botulism [67]. Previously, Woodburn et al. [43] recommended a minimum heat treatment of twenty minutes at 79 °C or 5 min at 85 °C for the inactivation of 10^3^ LD_50_ botulinum toxins (A, B, E, and F) per gram of the foods and buffer tested (0.05 phosphate buffer pH 6.2, 0.05M acetate buffer pH 4.2). Overall, all types of BoTNs are inactivated after 10 min at 100 °C or 30 min at 80 °C [22,68,69]. However, the time for the total inactivation of BoNT at lower temperatures is so long that it is impractical in food processing systems [70].

Thermal treatments in combination with other physical treatments of food can have synergistic or additive effects in managing the botulinum hazard. These combinations will be discussed in the following treatment methods.

## 4. Ionizing Radiations to Inactivate *C. botulinum*

Ionizing radiations are electromagnetic or corpuscular radiations having sufficient energy to detach electrons (ionize) from atoms and molecules coming in contact with them [71]. Irradiation effects are achieved without a significant increase in the temperature of the food product, thus, conserving the heat-sensitive components of food [4].

Mainly the β (corpuscular) and γ (electromagnetic) radiations are used for food with phytosanitary treatment (e.g., disinsectization), pasteurization, or sterilization objectives. The irradiation treatment process is measured in terms of the absorbed dose or energy per unit mass, and the standard unit of absorbed dose is the kilogray (kGy). The dose limits are based on the food product type, specific objective (phytosanitary treatment, pasteurization, or sterilization), and regulatory limits [4].

The β irradiations (electron beams, e-beams) are categorized as low-energy (< 1 MeV), medium-energy (1–8 MeV), and high-energy (8–10 MeV) applications. The maximum allowed limit for food products is 10 MeV, and at this energy level, the dose absorbed is ~3 kGy/s. These rays have low penetration (3.5 cm/density at 10 MeV) and they are reserved for homogeneous and thin product treatments. Electron beam accelerators are used to obtain the e-beams. These accelerators have an electron source (cathode), an accelerator tube, and a beam-shaping system. When the electrical excitation is turned on, the cathode produces a flow of electrons that is accelerated. Therefore, these characteristics make the system suitable for direct application in the food industry [4].

Gamma rays applied to food products are generated in specialized facilities (basic nuclear installation) by the decay of the radioisotope cobalt (^60^Co) or cesium (^137^Cs). The ^60^Co emits two photons with an average energy of 1.25 MeV with a high penetration capability [71]. The radioactive material is kept inside the double-layered stainless-steel coverings and immersed in a deep pool of water. For the treatment, the radioactive source is taken out of the pool at the level of the batch of packaged feed or foods previously installed in the treatment zone [4].

The β and γ radiations cause ionization of the products by detaching electrons from the atoms. Therefore, their mode of action and the result of microbial destruction are also the same [72]. They cause direct damage to cell components such as carbohydrates, proteins, DNA, and lipids. Meanwhile, the water molecules present in the food are hydrolyzed to produce free radicals and reactive oxygen species [71]. These short-lived and highly reactive entities react with the biological molecules in the immediate vicinity and cause damage to the vegetative microbial cells. Ionizing radiation also causes the inactivation of the spores and the effects can be seen in the form of structural damage, spilled cytoplasmic contents, reduced membrane integrity, and fragmented genomic DNA [73].

The resistance of spores to irradiation depends upon multiple factors, such as the type of spores (e.g., spores of proteolytic or nonproteolytic strains), the temperature of the product and environment, the medium of treatment, and the initial dose of microorganisms [74,75] (Table 4). The proteolytic *C. botulinum* spores are more resistant to irradiation as compared to nonproteolytic strains. An irradiation dose of 2.0–4.5 kGy is required to achieve a one-log reduction in spores for proteolytic and 1–2 kGy for non-proteolytic *C. botulinum* strains in frozen foods [22]. The dose necessary for 12-log reduction or the minimal radiation dose can be established based on these D-values, but it varies depending upon the food matrix, initial microbial contamination/inoculation dose, bacterium type, and environmental conditions [76,77,78,79,80]. However, the maximum irradiation doses have an upper limit according to regulations. For example, a maximum irradiation dose of 10 kGy is allowed for dried aromatic herbs, spices, and vegetable seasonings [81]; however, this dose limit is < 30 kGy when the said products are used as food ingredients in small quantities [82].

Food matrix can have a protective effect on spores, and therefore, a stronger dose is required for the inactivation of spores in different solid foods compared to a spore suspension in water or buffers [76,87,92] (Table 4). Similar is the case when the food is frozen (solid) or in liquid form. Erlinda [93] observed that a 10 kGy dose decreased the *C. botulinum* spores by 5 decimals in liquid while only 4 decimals in frozen food. Likewise, an approximately 9 kGy higher gamma radiation dose was required to inactivate the *C. botulinum* 33A spores in the cans of ground beef at −196 °C than at 0 °C [75]. Grecz et al. [18] later reported that the physical blocking by the frozen medium, plus radical scavenging by pork pea broth eliminated the indirect effects of the radiation.

Treatment temperature also influences the sensitivity of spores to irradiation. Grecz et al. [18] studied gamma irradiation of cooked beef packs at 14 temperatures ranging from −196 °C to 95 °C. They observed that the resistance of type 33A spores progressively decreased with increasing temperature, and the radiation death was much more rapid above 65 °C [18]. Anellis et al. [78] also observed that the resistance of the strain 33A spores decreased with increasing the temperature of the phosphate buffer solution from −196 °C to 5 °C.

Different types of suspending media can positively or negatively contribute to the radiotolerance of *C. botulinum.* Most foods and microbiological media are radical scavengers, and some of their additives also have similar properties. Lim et al. [83] reported that Na_2_–EDTA (0.01 M) was the most efficient radioprotector of *C. botulinum* spores due to its reactivity toward hydroxy radicals, followed by *t*-butanol (0.1 M) in NO_2_ or N_2_-saturated buffers, respectively. Huhtanen [89] observed that irradiation D-values of *C. botulinum* spores in honey were very high (8–13 kGy) compared to the water solution (2 kGy); the difference is believed to have occurred because of the presence of radio-protective free radical scavengers such as ascorbic acid, fumarate, or glutamate in the honey. However, at a 25 kGy dose, the sterilization of honey was achieved, thereby eliminating approximately 10^6^ spores of *C. botulinum* [90]. On the other hand, some media may promote the formation of harmful radicals, e.g., phosphate buffer forms phosphinic acid, and oxygen forms peroxy and perhydroxy radicals [75], thus decreasing the irradiation resistance of spores.

Radiation exposure affects both the cells and the spores through their membrane damage. This effect can render the cells more susceptible to other chemical stresses, such as NaCl [61]. Lim et al. [83] reported a reduction of viable spore counts in the γ-irradiated NaCl medium. As some meat products are salted, the irradiation can be synergistic with salting in increasing the shelf life. Barbut et al. [94] observed that medium-dose gamma irradiation (5 kGy) at two temperatures (1 °C and −30 °C), in turkey frankfurters containing at least 2.5% NaCl levels was sufficient to inhibit BoNT production for 40 days. However, the toxin was produced at lower doses of NaCl, indicating that medium-dose gamma radiation treatment cannot compensate for NaCl reduction in meat products. Therefore, the risk of spore germination and toxigenesis persists when an insufficient irradiation dose is applied, especially when the product is not stored in the cold [86,95,96,97,98].

During the treatment, the presence of some gasses in the medium can support the ionization process and result in higher sensitivity of microbes. Lim et al. [83] observed that the presence of O_2_, N_2_O, and N_2_ gases in the medium sensitized the spores to irradiation, probably because these environments supported the production of free radicals and reactive oxygen species. Contrarily, during storage, the absence of air, such as in vacuum packaging, provides suitable anaerobic conditions for *C. botulinum* [68]. Meanwhile, supplementation with oxygen has two effects: first, the aerobic bacteria can scavenge the oxygen, promoting the growth of Clostridia [68]. Second, the supplementation with oxygen alone can support the formation of CO_2_, which can stimulate spore germination and toxin production in *C. botulinum*. Despite the packaging conditions, gamma-irradiated food still provides a longer shelf life compared to non-irradiated food in similar storage conditions [97,98].

Gamma irradiation can also reduce the effect of some chemicals added as food preservatives. For example, irradiation causes a dose-dependent reduction in the quantities of sodium nitrite (NaNO_2_) in meat products, subsequently, resulting in a loss of anti-botulinal activity [99]. However, as the irradiation does not completely eliminate the nitrite quantities, the anti-botulinal effects still persist for a long time [6,100]. For example, Barbut et al. [94] reported that 40 mg/kg of sodium nitrite and 15 kGy sufficiently prevented toxin formation in *C. botulinum* (spores types A and B/g) inoculated bacon during two months of storage at 27 °C.

Irradiation may kill desired bacteria in food. For example, Huhtanen et al. [101] reported that low-dose irradiation (1.9 kGy) prevented acid production in the low-nitrite comminuted bacon. The probable reason was that the irradiations inactivated the acid-producing bacteria, and as a result, more rapid toxin production of *C. botulinum* was observed. However, at higher doses of ^137^Cs irradiation (3.8, 7.5, 11.2, and 15 kGy), protective effects were obtained, because these doses efficiently inactivated both the acid-producing bacteria and the *C. botulinum* spores.

The combination of ionizing irradiation with other forms of physical treatments can have complementary or synergistic effects on the sensitivity of *C. botulinum* cells [18]. It was reported that the combination of thermal treatment and γ irradiation improved the sterilization of canned peas and ground beef as compared to a single treatment [88,102,103]. Heating before irradiation increases the sensitivity of *C. botulinum* vegetative cells. However, in the case of spores, the sensitivity does not always increase with heating. There are two possible explanations for this difference: (i) sublethal heating activates the repair mechanism of spores, making them resistant to low-dose irradiation treatment; (ii) heating causes the coagulation of proteins and constriction of fibers (in the case of meat), and the spores hidden inside may be protected from irradiation treatment [16]. Contrarily, the irradiation of spores makes them more sensitive to heat, which might be due to the difference in the mode of action between ionizing radiation and heating [16,17]. 

The recovery of irradiation-injured spores depends upon multiple factors including the temperature and composition of the medium [104]. Chowdhury et al. [61] observed that the radiation-injured strain 62A spores did not germinate to form colonies at 50 °C, but, they germinated and repaired better at 40 °C compared to 30 °C. The UV radiation of *C. botulinum* spores also makes them more sensitive to gamma irradiation, thereby reducing the D-values [105].

The BoNT is inactivated by ionizing radiations depending upon the irradiation dose, type of toxin, temperature, and medium of treatment. Rose, et al. [106] reported that an 8 kGy dose inactivated the BoNT in the gelatin phosphate buffer. However, at the same dose, about half of the initial BoNT quantity remained active in minced beef slurries. Moreover, the toxin persisted at 15% in meat slurry even at 24 kGy [106]. Early studies on the subject observed that the BoNT was formed in the food even when sterilized using high doses of irradiation. Kempe and Graikoski [91] reported that the 33–38 kGy irradiation dose was sufficient to sterilize the cooked and raw ground meat in cans inoculated with *C. botulinum* 213B and 62A spores at different concentrations. No evidence of outgrowth was found in any of the cans, and spores did not germinate when tested for growth on media. Interestingly, the cans inoculated with 62A, at greater than 2.7 × 10^6^ spores/g of meat, showed the presence of toxin (tested by injecting the mice). Which meant that the toxin was formed even in the absence of spore germination. Thus, it appeared that the nonviable spores were able to produce a significant level of BoNT. Meanwhile, Costilow, [107] observed that ionizing radiation levels of 12.5 kGy were sufficient to render the *C. botulinum* 62-A spores nonviable (killed) but did not significantly affect the basic levels of the enzymes required for their primary catabolic processes. Later, another study by Grecz et al. [108] reported that such levels of BoNT might have originated from inside the spores. They described that the spores, even when thermally inactivated, somehow protected the toxin and then the toxin might have passed the spore membrane pores during the storage period. Overall, although a high irradiation dose can make the food sterile, it may take up to 15 h for this treatment depending upon the source and method of irradiation. In this situation, if the irradiation is happening at cold temperatures, there will not be any microbial activity. However, if irradiation is performed at warmer temperatures (e.g., 10–38 °C), during this long treatment, spores may start stress response germination and toxigenesis, as happens at sublethal doses [109]. This might be the reason that the aforementioned studies have reported the presence of BoNT in gamma-sterilized food [91,91,107,108,109]. Fernandez et al. [110] studied the toxicity of strain 33A spores in canned ground beef irradiated to 45 kGy at temperatures of −25, 0, and 25 °C; they found small minimal lethal doses of toxin in the irradiated samples which could be due to (i) the germination of irradiated spores, (ii) slow lysis of spores at 25°, (iii) new toxin synthesis by those spore enzymes which remained active after irradiation, (iv) activation or fragmentation of pre-existing toxin molecules, etc. Meanwhile, the irradiation in combination with high temperatures (i.e., >40 °C) can synergistically inactivate the BoNT [17]. Therefore, it is advised to use a combination of treatments to eliminate BoNT and strictly ensure good manufacturing practices in order to prevent the formation of toxins.

## 5. Using High Hydrostatic Pressure (HHP) for *C. botulinum* Inactivation 

High-pressure processing (HPP) or high hydrostatic pressure (HHP) is a nonthermal method used to inactivate food spoilage and pathogenic microbes in food and to modify the physicochemical nature of the food matrix [111]. In this method, high pressure (100 to 1000 MPa) is applied to packaged food (with sufficient water content and no air voids) at a specific temperature (0 and 120 °C) for a few seconds to several minutes [112,113]. This technique uniformly applies pressure on an isostatic food matrix, therefore, the effects are the same in the product irrespective of the shape, mass, volume, and physical state of the food [114].

The HPP affects noncovalent bonds, such as ionic, hydrophobic, and hydrogen bonds. Even though the primary protein structure remains intact, the changes occur in the secondary, tertiary, and quaternary structures, which consequently results in protein denaturation, aggregation, or gelation [111,114]. This technique may also influence carbohydrates by breaking the low-energy bonds of sugars, resulting in starch swelling and gelatinization. Crystallization occurs in lipids, which changes the permeability of membranes. Overall, the modification of the physicochemical characteristics and functional activities of biomacromolecules results in the inactivation of cells [111].

The organism-related factors (i.e., type of bacterium, growth phase, inoculum dose, stress/injury status, optimal growth temperature), food matrix (i.e., pH, water activity, physicochemical composition), and the treatment conditions (i.e., temperatures, air, packaging, pressure fluid) influence the resistance of cells and spores to the applied HPP or even to the combination of treatments [15,19,62,115] (Table 5). Likewise, the recovery of spores after heating and HPP application depends upon the media, additives, and incubation conditions [35,116].

The *C. botulinum* spores are very resistant to HPP at ambient temperatures [125]. Even a very high pressure (1500 MPa) at 20 °C for 5 min did not significantly inactivate the spores [126]. Similar resistance was observed when the temperature was increased to 35 °C at 827 MPa for 5 min [117]. In addition, the inactivation curves also show a pronounced pressure-dependent tailing, indicating the survival of a small fraction of the spores up to 120 °C and 1400 MPa in isothermal treatments [127]. Given this problem, the HPP-treated foods stored at room temperature should be considered at risk of growth and toxin production by *C. botulinum*.

Combined HPP/HT (high temperature) treatments are better solutions to inactivate the spores compared to heating or HPP alone [13] (Table 5). A 2–3-log reduction of *C. botulinum* spores (type BS-A and 62-A) in the phosphate buffer (0.07 M, pH 7.0) and crabmeat blend have been obtained using 827 MPa at 75 °C for 20 min [118]. Sterilization was achieved for the 5-log spores/mL suspended in a liquid media (meat and carrot broths) when treated with >800 MPa for 5 min at initial temperatures of 80 to 90 °C [126]. By increasing the pressure and temperature simultaneously, the efficacy of HPP increases, but it is not the same for all types of *C. botulinum* [119]. Reddy et al. [128] reported that by increasing the pressure from 600 to 750 MPa at 105 °C, D-values of some *C. botulinum* strain spores decreased (i.e., for 69-A, 1.91 to 1.33 min and for PA3679, 2.35 to 1.29 min). In another study, the researchers observed that at higher temperatures (117–121 °C), increasing the pressure from 600 to 750 MPa decreased the pressure-assisted thermal D-values of PA3679 (from 0.55 to 0.28 min) but not of Giorgio-A and 69-A strains [129]. 

It is also important to note that the heat resistance of spores is not correlated with baroresistance [120]. Shao et al. [123] studied the high-pressure destruction kinetics of *C. botulinum* (Group I, strain PA9508B) spores in milk at high temperatures. They observed that during HHP/HT treatment, the spores were relatively more resistant at higher pressures than at higher temperatures. Furthermore, the D-value trends beyond 120 °C indicated that heating was the principal mode of spore inactivation, while the high pressure protected them under these conditions [123].

Combining irradiation with HPP has also been studied, but it has not been practiced very much in the field. Crawford et al. [14] observed that the pretreatment of the chicken breasts with HPP reduced the irradiation D-value to approximately 2 kGy.

The inactivation of spores with HPP also depends upon the bacterial strain and the food matrix [15,19] (Table 5). Generally, the HPP and temperature resistance of different *C. botulinum* is: proteolytic types A, B > nonproteolytic type B > (sensitive type B strains) ≥ non-proteolytic type E [60], however, a large variation can exist in each category. A 5-log reduction of the proteolytic type A *C. botulinum* strain was recorded using 600 MPa at 80 °C for 12 min. The same treatment (600 MPa at 80 °C) could reduce only 3 logs of the proteolytic type B strain even after 60 min [120]. Bull et al. [15] observed that heating and HPP acted synergistically for *C. botulinum* FRRB 2802 (NCTC 7273) and *C. botulinum* FRRB 2804 (NCTC 3805 and 62A) in the Bolognese and cream sauces and for *C. botulinum* FRRB 2807 (213B) in the Bolognese sauce only. On the other hand, no such synergistic action was reported for *C. botulinum* FRRB 2803 (NCTC 2916) or FRRB 2806 (62A), or *C. sporogenes* FRRB 2790 (NCTC 8594 and PA3679) in any of the model products. 

The food packaging material can also influence the HPP treatment efficacy. Patazca et al. [19] reported a 6.5-log reduction of the spores packaged in the graduated tube or cryovials after processing for up to 10 min at 118 °C and 700 MPa. On the other hand, only a <4.8-log reduction was obtained for the spores packaged in plastic pouches. In addition, the permeability of packaging to oxygen can also influence the germination of *C. botulinum.* Linton et al. [130] studied the germination of five non-proteolytic *C. botulinum* strains inoculated into raw chicken mince that was cooked and then pressure treated at 600 MPa for 2 min at 20 °C. The spores not only survived the treatments but also germinated during storage. However, the germination and growth were controlled when 2% *w/w* sodium lactate was added and when oxygen-permeable packaging was used.

## 6. Emerging Non-Thermal Technologies and Their Potential Application to Reduce *C. botulinum* Hazard

Recent innovations in food processing energy are working towards improving food safety without alteration of the quality of the food product. Therefore, more and more non-thermal treatments are being investigated to improve food safety. The following section will discuss the potential of such technologies for controlling the *C. botulinum* hazard in the food industry. However, the majority of these techniques have either not at all or not sufficiently been assessed regarding the control of *C. botulinum* hazards.

### 6.1. Pulsed Electric Fields (PEF)

As the name indicates, this method applies a pulse of high field intensity (25 to 85 kV/cm) to food for a very short duration of a few milliseconds or nanoseconds [131]. This rapid method does not increase the temperature of food and thus conserves organoleptic qualities [132,133].

The high field intensity directly damages the membranes of microbial cells [134,135]. It also results in oxidative changes in the lipids and proteins of the cells, leading to the inactivation of the metabolic enzymes [131]. Pillet et al. [136] reported that the PEF caused structural disorganization correlated with morphological and mechanical alterations of the cell wall in the vegetative cells of *Bacillus pumilus* (a non-pathogenic model of food contaminants like *C. difficile*, *C. botulinum*, *B. cereus*). They also reported that the PEF caused a partial destruction of coat protein nanostructures, which is associated with internal alterations of the cortex and core of the spores [134,135,136].

The efficiency of PEF in reducing the microbial load in food depends upon various intrinsic and extrinsic factors related to methodology, microbe, and the food matrix. The methodology of PEF involves the intensity of the field applied, the total exposure time, temperature, and energy [131]. All of these cumulative factors increase the efficiency of PEF to inactivate the vegetative cells and spores. Pillet et al. [136] studied the effect of PEF strength on the inactivation of *B. pumilus* cells suspended in water by applying 1000 pulses of 5 μs from 2 to 7.5 kV/cm. The inactivation of cells varied from 38 to 98% between 2 and 6 kV/cm and was partially due to reversible permeabilization. Meanwhile, the strongest electric field (7.5 kV/cm) inactivated the cells entirely by irreversible permeabilization. In contrast, these conditions (1000 pulses at 7.5 kV/cm) were unable to inactivate the spores. However, increasing the number of pulses to 10,000 inactivated 67 ± 8% spores. Rezaeimotlagh et al. [137] showed that processing with low frequency at 12.5 kV/cm and 40 °C, followed by a high-frequency electric field at 2.1 kV/cm and 65 °C, had a synergistic effect on *Escherichia coli* by 5 logs in saline water and cranberry juice. In such studies, the conductivity of the medium can influence the aforementioned factors, which means that if there is low conductivity of the medium, the lethal dose would not be obtained by increasing the intensity of PEF [134]. The conductivity of the medium depends upon the food matrix, pH, aw, and chemical composition. Moreover, the microbial factors, including the types of microbes, their growth and germination stage, and stress/injury level also influence the efficacy of PEF [134]. Qiu et al., [135] reported that spores treated with nonlethal PEF doses respond to germinants more quickly and with less heterogeneity, possibly because the tiny cracks formed on the spore membranes facilitate the germinants’ access to the germination receptors on the inner membrane. 

Overall, PEF treatment can be an alternative to food treatment to inactivate the vegetative cells [138]; however, it is generally insufficient to inactivate bacterial spores [135,139]. Meanwhile, decreasing spore resistance through germination followed by parameter-optimized PEF treatments has been reported as a method for improving pasteurization efficiency [134]. Thus, it is likely that this efficiency in inactivating *C. botulinum* spores in different food matrixes can be increased by combining different physical treatments [138]. For example, Siemer et al. [140] reported a combined application of PEF and thermal energy to achieve higher electric field strengths with less inactivation energy, even at lower conductivity of the medium, which ultimately resulted in higher inactivation of *B. subtilis* spores. Gomez–Gomez et al. [138] reported a synergistic inactivation effect of combined PEF and high-power ultrasound treatments on *B. pumilus* cells and spores in the oil-in-water emulsions. Soni et al. [139] observed that the PEF treatment at 9.4 kV/cm at 80 °C (pulse width of 20 μs and frequency of 300 Hz) caused a 3-log reduction of *B. cereus* spores and importantly, the D_88 °C_ values of the surviving spores were reduced by 12 min.

### 6.2. Intense Light Pulses (ILP)

This approach involves delivering high-intensity light radiation in the form of intense, intermittent, short-duration pulses. This high-intensity light includes a wide wavelength ranging from 200 to 1100 nm, including UV (200–400 nm), visible (400–700 nm), and near-infrared region (700–1100 nm) radiation waves [133]. As the penetration of light through opaque objects is limited, this technology is currently applicable for treating surfaces and transparent fluids [141,142].

The main microbial inactivation by ILP is due to UV radiation which causes DNA structural changes and membrane damage by affecting the conjugated carbon–carbon double bonds in nucleic acids [133]. Moreover, ILP has been reported to denature the cell wall structures, cause cytoplasm shrinkage, and rupture of the internal organization leading to leakage of cytoplasmic content and ultimately to cell death [141]. The susceptibility trend is reported to be Gram-negative bacteria > Gram-positive bacteria > bacterial spores > fungal spores (reviewed in [133]).

Like other physical treatments, various intrinsic and extrinsic factors influence the efficacy of ILP for inactivating different microorganisms. The microbe-related factors are the type of microbe, the growth stage, live forms (vegetative vs. spores), the inoculation dose, etc. [143,144]. Dittrich et al. [145] reported that ILP fluences (9.8–13.3 J/cm^2^) significantly reduced the *Salmonella* spp. load in dried parsley by 0.3–5.2 logs depending upon the isolate (*S.* Cerro and *S*. Agona), cell density, and storage treatment. Cassar et al. [144] reported that the *B. cereus* endospores were more resistant to different ILPs compared to vegetative cells. Jo et al. [143] reported that the ILP fluence of 7.40 J/cm^2^ resulted in a 7-log reduction *B. subtilis* spores. The resistance of spores to ILP increased during the initial germination period and then decreased at subsequent stages. This temporary increase in resistance was attributed to the leakage of DPA from the spores.

The food matrix-related factors also influence the efficacy of ILP against bacterial cells and spores. Huang and Chen [146] studied the use of water-assisted ILP on spot- and dip-inoculated *Salmonella* spp. in fresh produce. They observed that the ILP treatment reduced the bacterial load on blueberries, tomatoes, and lettuce shreds by 4.5, 4.4, and 2 logs, respectively, on the spot-inoculated fresh produce, whereas only about 2 logs were reduced for all the dip-inoculated samples. Hwang et al. [142] used ILP treatment conditions of a lamp DC voltage of 1.8–4.2 kV, a pulse width of 0.5–1.0 ms, a frequency of 2 Hz, and a treatment time of 1–5 min for four powder foods. Under a total energy fluence of 12.31 J/cm^2^, the total mesophilic aerobic bacteria reductions of 0.45, 0.66, 0.88, and 3 logs CFU/mL were recorded for ground black pepper, red pepper, embryo buds of rice, and sesame seeds, respectively.

The technology-related conditions also influence the ILP treatment efficacy [144]. Hwang et al. [146] reported that the distance between the sample and the IPL lamp (8, 13, and 18 cm), the pulse width (0.5, 1.3, and 2.1 ms), the charging voltage (1, 1.2, and 1.4 kV), and the processing time (10, 20, and 30 s) all considerably influenced the *B. subtilis* (KCCM 11,315) spore inactivation rates. In their study, the optimal treatment conditions producing a 6-log reduction were a distance of 8 cm, a pulse width of 2.1 ms, a charging voltage of 1 kV, and a processing time of 30 s. It is important to mention that the lethal effect of charging voltage was more evident when the distance was 18 cm. Therefore, it is very important to identify the optimal combinations of voltage, pulse width, and fluence in order to attain maximal microbial inactivation. Bousi et al. [147] reported that the higher voltages delivered more energy with pulse width (i.e., a higher irradiance) resulting in a more lethal process. The treatment was very effective for a 6 to 7-log reduction for four different *Salmonella* serovars and their cocktail on the medium surface, after a single light pulse with a fluence of 338 mJ/cm^2^ and 280 mJ/cm^2^, at 2.5 kV (200 μs) and 3.0 kV (100 μs), respectively. More intense ILP is recommended in order to attain different safety criteria, e.g., pasteurization. Bhagat and Chakraborty [132] reported that 761.4 J/cm^2^ (2.7 kV for 90 s) pulsed light treatment was required for attaining the 5-log reduction (microbial safety) in *E. coli* ATCC 43888, aerobic mesophiles, yeasts, and molds count in the pomegranate juice, which was similar to the thermal treatment attaining the equivalent lethality at 95 °C for 2 min. Moreover, 3 kJ/cm^2^ pulsed light and a thermal treatment of 95 °C for 3 min fully inactivated enzymes (polyphenol oxidase and peroxidase). Thermal treatment, however, caused the depletion of phenolic and antioxidants compounds, while the intense pulse dose of 3 kJ/cm^2^ retained higher quantities of phenolics (97%), antioxidants (94%), and ascorbic acid (83%) [132]. These encouraging results show that ILP combined with thermal treatments should be explored regarding the inactivation of bacterial toxins including BoNT.

### 6.3. Cold Plasma (CP)

Plasma is a partially ionized gas that constitutes the fourth state of matter. It contains highly reactive species such as ions, radicals, electrons, photons, and excited molecules. This ionization is achieved by passing a carrier gas (air, oxygen, nitrogen, helium, argon, etc.) through an electric field generated by various means, such as microwaves, pulses, A.C., D.C. electric fields, etc.) [148]. The reactive species of CP can disrupt the proteins, fats, and carbohydrates, leading to the inactivation of microbial cells at room temperature [149,150,151,152,153].

Cold plasma has been used for 5-log reduction of various microbes of food safety concern, e.g., *Salmonella* spp., *E. coli* O157:H7, *Listeria monocytogenes*, and *Staphylococcus aureus* [149,150]. The D-values, for the inactivation of *E. coli* ATCC8739 and *B. subtilis* ATCC6633 under gas discharge plasma treatment, were reported to be just about 30 s [154].

CP has also been reported to effectively inactivate bacterial spores. Dobrynin et al. [155] reported that atmospheric-pressure dielectric-barrier-discharge plasma treatment at a discharge power of 0.3 W/cm^2^ effectively inactivated *B. cereus* and *B. anthracis* spores on a solid surface and in plastic and paper packings within a minute. They described that the neutral reactive oxygen species and UV radiation played a dominant role in the inactivation of spores. Sorto [156] studied the inactivation of *B. atrophaeus* spores using CP treatment. The results showed a 1.8-log CFU/mL reduction in spores after 6 min of CP exposure (60 kV).

Generally, it has been proposed that CP inactivates bacterial spores by targeting the coat/inner membrane [154], DNA, and metabolic proteins of spores [153,157]. Tseng et al. [154] reported that the helium atmospheric CP jet effectively inactivated the spores of various bacteria, and the D-values were highest for *C. botulinum* type A ATCC3502 (8.04) followed by *C. sporogenes* ATCC 3584 (5.27), *Geobacillus stearothermophilus* ATCC7953 (4.72, the bacterium formerly known as *Bacillus stearothermophilus*), *C. botulinum* type E NCTC11219 (3.5), *B. subtilis* ATCC6633 (3.5), *C. difficile* 6871 (2.8), and *C. perfringens* ATCC3624 (2.7). According to the authors, such high resistance of *C. botulinum* type A spores might have been due to some unique protective structures, e.g., spore coats, inner membranes, and the small, acid-soluble proteins (SASP), which increased the spore resistance to plasma and radiation. 

Similarly to the aforementioned treatments, the bacterial cell and spore inactivation ability of CP treatment depends upon process parameters (electrode type, input voltage, time, dose, sample distance, gas type, gas ratio, etc.) [148,152,153,158], food matrix (composition, form, pH, aw) [151,153,159], and microbial factors (type, growth stage, vegetative or spore form, etc.) [151,153,154,156,160]. 

Globally, each technology has its pros and cons based on the perspective of employment and the objectives to be achieved. The availability and cost of technology vary in different parts of the world; therefore, it is very difficult to summarize the economic aspect. Potential advantages and disadvantages of these physical treatments have been listed in Table 6. Overall, this article mainly discussed the suitability of these technologies in terms of their potential to control *C. botulinum* hazards.

## 7. Conclusions

Thermal treatment remains the main method of controlling the *C. botulinum* hazards in food. To achieve commercial sterility through 12-log reduction in low-acid canned foods, thermal treatment or ‘botulinum cook’ is the main method of choice. The thermal treatments, even with a short duration of application, can change the organoleptic quality of the product. Therefore, adapting non-thermal technologies or combining them with thermal treatments can preserve the nutritional and sensory properties of the products and meet the food safety objectives.

Irradiation is successfully used in combination with heat treatment to inactivate *C. botulinum* in different food products. The availability of the technology and customer acceptance are the main limitations in adapting irradiation for food.

*C. botulinum* spores are generally resistant to HPP. However, the combination of thermal and HPP treatments has shown promising results in effectively inactivating spores in different food products. As a non-thermal alternative, the combination of HPP and irradiation can also be used to exert synergistic effects for inactivating *C. botulinum*.

Pulsed electric fields, intense light pulses, cold plasma, and other emerging technologies are still new to the food safety industry. There is limited research on the ability of these technologies to control *C. botulinum* in food. Therefore, one way to move forward is to research the application of these technologies on model or surrogate bacteria which will ultimately increase our confidence in the extrapolation of these results to *C. botulinum*.

## Figures and Tables

**Figure 1 foods-12-01580-f001:**
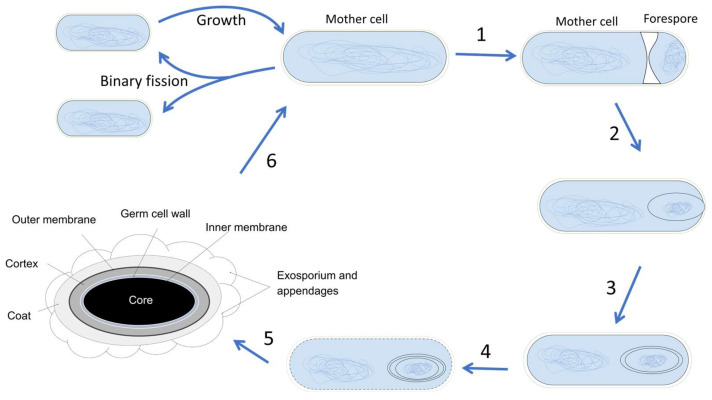
Structure of *Clostridium botulinum* spore and the key stages of sporulation: (1) Asymmetric division: vegetative cells normally divide by binary fission, and under stress conditions, they start sporulation process by initiating asymmetric division. (2) Engulfment: the newly emerged forespore is engulfed by the parent cell. (3,4) Cortex and coat formation: the cortex is formed around the forespore, and the coat starts depositing from mother cell membranes. (5) After maturation of the endospore, mother cell lysis occurs, releasing a dormant spore (structure shown with labels). (6) Germination: the spore germinates into a new bacterial cell under favorable conditions.

**Table 1 foods-12-01580-t001:** Classification and characteristics of principal *Clostridium botulinum* strains and their toxins *.

Group	I	II	III
Toxins type	A, B, F	B, E, F	C, D
Toxins sub-types	A1, A2, A3, A4, A5, A6, A7, A8, B1, B2, B3, B5, B6, B7, B8, bivalent B (Ba, Bf, Ab),F1, F2, F3, F4, F5, F8, X	E1, E2, E3, E6, E7, E8, E9, E10, E11, E12, B4 or non-proteolytic B, F6 or non-proteolytic F	C, C/D, D, D/C
Proteolysis	Yes	No	No
Toxins genes support	Chromosomic/Plasmidic	Chromosomic/Plasmidic	Bacteriophage
Related non-toxigenic bacteria	*C. sporogenes*	*C. taeniosporum*	*C. novyi* *C. haemolyticum*
Saccharolytic activity	-	+	-

* Adapted from ANSES [3].

**Table 3 foods-12-01580-t003:** Growth, survival, and toxin production characteristics of *C. botulinum* strains implicated in food safety *.

Characteristic	Environment	Group I	Group II
Vegetative cell growth	Temperature opt (min-max) °C	35–40 (10–48)	18–25 (1.5–45)
	pH opt (min–max)	7.0 (4.6–9.0)	7.0 (5.0–9.0)
	Aw (min) with NaCl	0.94	0.97
	Aw (min) with glycerol	0.93	0.94
	NaCl % preventing the growth	≥10	≥5
Toxin production	At min temp	Yes	Yes
	At min aw	Yes	Yes
Spore resistance	Heat	D 120 °C = 0.04–0.72	D 80 °C = 0.23–2.63
		Z ~ 10 °C	
	Freezing resistance	Yes	Yes
	Irradiation D-value (T °C ≤ −18) **	2.0–4.5 kGy	1.0–2.0 kGy
Toxin stability	Heat	Denatured after 10 min at 100 °C or 30 min at 80 °C	
	Freezing	Stable at freezing even after 3 thawings and refreezings	
	pH	More stable in stronger acidic conditions ***	

Min = minimum; max = maximum; * Adapted from the literature data [22,24,43,44,45,46,47,48]; ** Frozen food. *** In general, the toxins were more stable in acid foods such as tomato soup at pH 4.2 than in low-acid foods, such as canned corn at pH 6.2 [43].

**Table 4 foods-12-01580-t004:** Ionizing radiation doses for the inactivation of *C. botulinum* strains in different food matrices.

*C. botulinum* Strain	D-Value (kGy *)	Medium/Substrate	Temperature °C	Reference
Type E Eklund	1.8	Sodium phosphate buffers	24	[83]
Type E Beluga	3	Vacuum-sealed chicken skins	5	[84]
Type A	2.7–3.2 (1.8 for green beans)	Chicken parts, beef steak, pork loin, green beans	30	[85]
Type E (VH, Beluga, 8E, 1340E, Iwanai, Alaska) A B	1.3 (1.28, 1.36, 1.38, 1.31, 1.25, 1.37) 2.8 2.4	Beef stew	30	[86]
Type A, type B, and non-toxic (102 strains)	3.3 (resistant strains); 2.4 (intermediate); 1.3 (sensitive)	Phosphate buffer	30	[74]
Type A (strain 33A)	2.9	Phosphate buffer, canned ground beef	0	[87]
	4.0		−196	
	4.6		0 °C	
	6.8		−196	
Type A (strain 33A)	3.3	Phosphate buffer	−196	[78]
	3.0		−140	
	2.5		−80	
	2.3		−30	
	2.0		5	
Type A (strain 33A)	5.8	Ground beef	−196	[18]
	5.6		−175	
	5.3		−140	
	5.1		−125	
	4.8		−100	
	4.6		−75	
	4.3		−50	
	4.1		−25	
	3.8		0	
	3.6		25	
	3.4		45	
	3.2		65	
	2.8		85	
	1.6		95	
PA 3679 **	3.4	Cooked,		[88]
	4.0	Raw meat		
Type A (strain 33A)	3.8	Ground cooked meat	0	[75]
Type E (VH, Beluga, 8E, Iwanai, Alaska, 16/63, Minneapolis, 1537/62, 4318/63)	1.5 (1.4, 1.2, 1.6, 1.2, 1.7, 2, 1.6, 1.5, 1.2)	Aqueous solution	Ambient temperature	[77]
Type A (strain 62A)	8–13	Honey	0–4	[89]
	2	Water		
ATCC 19397	3.7	Honey		[90]
Type 213B, Type 62A	3.7, 3.85	Raw and cooked ground meat	5	[91]

* kGy = kiloGray; ** *C. sporogenes* PA 3679 is widely used as a nontoxigenic surrogate for proteolytic *C. botulinum.*

**Table 5 foods-12-01580-t005:** Pressure, time, and temperature combinations to reduce the viability of *C. botulinum* spores in different media.

*C. botulinum* Type	Pressure (MPa)	Time (min)	Temperature (°C)	Log Reduction	Medium	Reference
Type E (Alaska and Beluga strains)	827	5	50–55	5	Phosphate buffer	[117]
		10	40			
Type BS-A and 62-A	827	20	75	2 and 3	Phosphate buffer	[118]
		15		3.2 and 2.7	Crabmeat blend	
Nonproteolytic type B (2-B, 17-B, KAP8-B, and KAP9-B)	827	20	75	6	Phosphate buffer and crabmeat	[119]
Proteolytic type B TMW 2.357	600	60	80	2	Mashed carrots	[120]
Proteolytic type B TMW 2.359				4		
Nonproteolytic type B (ATCC 25765 and TMW 2.518)		<1		5		
Type A–TMW2.299		60				
Type A–ATCC 19397		12				
Proteolytic type F		60				
Nonproteolytic type B, F, and E strains	600	6–40	80–91	5	N-(2-acetamido)-2-aminoethanesulfonic acid (ACES) buffer (0.05 M, pH 7.00)	[121]
	750	2–7				
Nonproteolytic type B, F, and E strains	600	7–9	80	1 *	N-(2-acetamido)-2-aminoethanesulfonic acid (ACES) buffer (0.05 M, pH 7.00)	[122]
	650	3–4				
	700	1.8				
Proteolytic PA9508B	700	21, 3.8, 0.6	90, 100, 110	1 *	Milk	[123]
	800	14, 2.7, 0.5				
	900	14, 1.8, 0.4				
62A	900	0.5/3 **	100	7/7	Phosphate buffer (0.1 M)	[124]
IB1-B				3.3/7		
CK2-A				1/4.5		
MRB				2.2/7		
Langeland				3.3/7		
A6				1/4.5		
GA0108BEC				1/4.5		
PA9508B				0/1.6		
13983B				1.7/7		
H461297F				1.5/6		
GA0101AJO				1.1/5		
HO9504A				1/3.3		
Type E TMW 2.990	600	10	90	6 *	Green peas with ham	[12]
		15			Steamed sole	
		11			Braised veal	
		10			Vegetable soup	

* Log reduction derived from the D-value studied; ** tested at two times: 0.5 and 3 min.

**Table 6 foods-12-01580-t006:** Advantages and disadvantages of different physical treatments regarding food quality, employability, and potential to control *C. botulinum* hazards.

Technology	Advantages	Disadvantages
Heating	A Traditional, mostly studied, and widely available technology Low initial cost Commercial sterility (12-log reduction) can be achieved at 121 °C within 3 min Inactivation of toxins, some allergens, and anti-nutritional factors Formation of desired flavor compounds	Loss of food freshness Nutritional loss (e.g., vitamins B and C) Rising energy costs Formation of undesired compounds (e.g., acrylamide, furans) Low heat pasteurization does not eliminate *C. botulinum* hazards
Ionizing radiations	Gamma radiation can be uniformly applied to thick food materials β radiations can be integrated on production line and turned on and off Does not change the nutritional and organoleptic properties of food Can be coupled with heating for synergistic and additive effects Can inactivate BoNTs using a very high dose (up to 60 kGy) Can achieve sterilization with a D-value of 1–4.5 kGy Maintains food freshness and diminishes sprouting	γ radiations cannot be integrated on production line The high installation cost for γ radiations (small nuclear facility) Difficulty in the procurement of radioactive materials γ irradiations are continuously emitted from the source and thus cannot be turned on and off Slow and low-dose processing may allow the formation/release of BoNTs Cost of transporting food to gamma irradiation facilities β radiations have low penetration into food Sterilization doses may be higher than the allowed dose in different countries Consumer reluctance to accept γ irradiated food Desired flavor compounds not formed
HPP/HHP	Preserves nutritional profile No formation of undesired compounds Retention of food freshness Can be coupled with thermal treatment for synergistic effects Can be coupled with non-thermal physical treatments for additive effects	Physical alteration of food Mainly affect the vegetative cells Desired flavor compounds not formed Anti-nutritional factors not inactivated
Emerging non-thermal technologies	Industrial interest in the innovation of modern technologies No formation of harmful compounds Preservation of freshness and physical characteristics of food	Not widely studied for *C. botulinum* spores Desired flavor compounds not formed Anti-nutritional factors not inactivated Sterilization is not easily achieved Require cold storage post-treatment

## Data Availability

Not applicable.

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
