# Peer review of "Physical Treatments to Control Clostridium botulinum Hazards in Food"

_foods, 2023, doi:10.3390/foods12081580_

Round 1

Reviewer 1 Report

Comments to Author:

The paper reviews the current knowledge on the suitability of the physical methods including heating, pressure, irradiation, and other emerging technologies to control C. botulinum hazards in food, which is comprehensive to guide the decision-makers, researchers, and  educators in using physical treatments to control C. botulinum hazards. To some degree, this review is instructive. However, I did have some concerns relative to the topic to be addressed, hence I provided some comments line by line. The detail of my considerations was presented below.

Topic concerns:

It is recommended to summarize the advantages and disadvantages of different physical methods to control C. botulinum hazards.

Topic edits:

Line 30: The initial letter should be written in the same case in the "Keywords" section.

There are too many sections in the "Conclusion" part, hence it is suggested to merge and rewrite them.

Language edits:

Line 21: "the food", please delete the word "the".

Line 78: "is divided in ": Replace "in" with "into".

Line 303: Please add the article "the" before the word "temperature".

Author Response

Topic concerns:

  • It is recommended to summarize the advantages and disadvantages of different physical methods to control C. botulinum hazards.

L 685: Summary table 6 added

Topic edits:

  • Line 30: The initial letter should be written in the same case in the "Keywords" section.

L30: Retained according to Foods template

  • There are too many sections in the "Conclusion" part, hence it is suggested to merge and rewrite them.

L 689-695: First two paragraphs merged in the updated manuscripts.

Language edits:

  • Line 21: "the food", please delete the word "the".

L21: Deleted “the”

  • Line 78: "is divided in ": Replace "in" with "into".

L78: Replaced "in" with "into".

  • Line 303: Please add the article "the" before the word "temperature".

L302-303: Added the article "the" before the word "temperature".

Reviewer 2 Report

The work is well structured and presents control alternatives for C. botulinum. I suggest expanding the discussion on sensory impacts on food with different treatments, as well as their costs and updated data on the occurrence of foodborne botulism in humans.

Author Response

L 685: Sensory aspects are added in the summary table. However, there is a limited information on the comparative costs. We have added the cost and organoleptic aspects in the Table 6 in the updated manuscript.

Moreover, it is difficult to compare the costs of old technologies, processing very large volumes of food with innovative technologies under development, whose equipment, both rare and new, is expensive today but whose cost should decrease in the coming years. For example, high pressure equipment that was worth around a million euros 20 years ago costs between 200 to 300,000 € today.

L 148-166: The occurrence of botulism has been discussed in the section ‘botulisms’. Rewriting of sentences done to highlight the importance in the section. 

Reviewer 3 Report

The article is interesting and constitutes a good review on this topic.

The abstract should be more specific in relation to the results obtained in this review.

The introduction should be an introductory text regarding the theme and the various points addressed, and only in the last paragraph should it present the objectives of the article. And the manuscript must be revise accordingly.

Although the review article is about physical treatments to control Clostridium botulinum, the introduction should mention that there are also other types of treatments for foodstuffs to avoid contamination by this microorganism, namely the use of specific food additives.

Tables should be formatted with the smallest letter and so that there are fewer breaks in the tables.

Some of the references are too old, and those that are not fundamental should not be cited. As very old references I highlight reference number 5 with 101 years, references numbers 13, 71, 75, 78, 79, 84, 85, 96, 97, 101, with more than 60 years, and even a significant number of other references about 40 or 50 years old.

Author Response

Section 6 discussed the food safety aspect of emerging technologies, and generally they have not been well studied for C. botulinum. We have added additional information (L 527-77, 595-597, 599-607, 655-661, 679-684, and table 6) of effect of emerging technologies on C. botulinum like microbes e.g. Bacillus spp.

Reviewer 4 Report

The subject is essential for the food industry, and an analysis of the most effective physical methods against C. botulinum is needed. The article presents essential aspects of C. botulinum, concerning the control and prevention of this bacterium in food, and “providing basic knowledge for educational purposes”. However, some recommendations are necessary:

Classification is necessary but unclear, the authors need to provide further clarification on the existing three or six proteolytic/non-proteolytic classes - lines 78-86.

In Table 1, in column three the corresponding line "Toxins sub-types" should be written in English - lines 95-96.

In table 2, the pieces of information should be better arranged because the colons limits are hard to be seen - lines 175-176.

The expression "the functioning of the nucleic acid" can be changed to "the function of nucleic acid", line 206.

Use "depending" instead of "depends", line 208.

 Is not very clear what means "by around 5 decimals, whereas in the frozen state, by 318 around 4 decimals", lines 318-319.

Concerning the methods used to inactivate C. botulinum, it is mentioned only once (lines 523-524) that the methods used do not alter food matrices. It is understood that only the technologies presented in section 6 fulfil the food safety criteria. Further clarification is needed!

Perhaps a table should also be drawn up for Section 6.

Author Response

  • Classification is necessary but unclear, the authors need to provide further clarification on the existing three or six proteolytic/non-proteolytic classes - lines 78-86.

L 78-86: There are 6 groups of neurotoxin - producing Clostridium (I-VI). The strains of the species C. botulinum basically belong to first 3 groups. Groups I and II are the main human pathogenic strains,  therefore the subject of this review.

  • In Table 1, in column three the corresponding line "Toxins sub-types" should be written in English - lines 95-96.

L 95: Corrected English

  • In table 2, the pieces of information should be better arranged because the colons limits are hard to be seen - lines 175-176.

L 175: Yes, this is MDPI template style, without column separations. We have made first column bold italic and added bullets to thirds column to avoid the confusions.

  • The expression "the functioning of the nucleic acid" can be changed to "the function of nucleic acid", line 206.

L 205: Corrected to ‘function’

  • Use "depending" instead of "depends", line 208.

L 207: Added “depending" instead of "depends", line

  • Is not very clear what means "by around 5 decimals, whereas in the frozen state, by 318 around 4 decimals", lines 318-319.

L 320-21 : Rewriting was done to avoid confusions

  • Concerning the methods used to inactivate C. botulinum, it is mentioned only once (lines 523-524) that the methods used do not alter food matrices. It is understood that only the technologies presented in section 6 fulfil the food safety criteria. Further clarification is needed!

Section 6 discussed the food safety aspect of emerging technologies, and generally they have not been well studied for C. botulinum. We have added additional information (L 527-77, 595-597, 599-607, 655-661, 679-684, and table 6) of effect of emerging technologies on C. botulinum like microbes e.g. Bacillus spp.

  • Perhaps a table should also be drawn up for Section 6.

L685 : Summary table 6 created

Round 2

Reviewer 2 Report

The article has had important improvements and can be published in its current form.

Author Response

Thank you for your consideration.

Reviewer 3 Report

There was certainly some error in the correction process, as the authors did not consider my correction requests, and cover Letter also does not measure them.

So I will repeat here my observations and correction requests:

The article is interesting and constitutes a good review on this topic.

The abstract should be more specific in relation to the results obtained in this review.

The introduction should be an introductory text regarding the theme and the various points addressed, and only in the last paragraph should it present the objectives of the article. And the manuscript must be revise accordingly.

Although the review article is about physical treatments to control Clostridium botulinum, the introduction should mention that there are also other types of treatments for foodstuffs to avoid contamination by this microorganism, namely the use of specific food additives.

Tables should be formatted with the smallest letter and so that there are fewer breaks in the tables.

Some of the references are too old, and those that are not fundamental should not be cited. As very old references I highlight reference number 5 with 101 years, references numbers 13, 71, 75, 78, 79, 84, 85, 96, 97, 101, with more than 60 years, and even a significant number of other references about 40 or 50 years old.

The numbering of references is slightly different, and now this list of very old references I sent in the first review may not correspond exactly to the actual numbering. However, the problem remains.
